

# Updated atmospheric mercury emissions from iron and steel production in China during 2000-2015

**Qingru Wu[1,2], Wei Gao[1,2], Shuxiao Wang[1,2*], Jiming Hao[1,2]**

[1]School of Environment, and State Key Joint Laboratory of Environment Simulation and Pollution Control, Tsinghua University, Beijing 100084, China

[2]State Environmental Protection Key Laboratory of Sources and Control of Air Pollution Complex, Beijing 100084, China

*Correspondence to:* S. X. Wang (shxwang@tsinghua.edu.cn)



**Abstract**

10       Iron and steel production (ISP) is one of the significant atmospheric Hg emission sources

in China. Atmospheric mercury (Hg) emissions from ISP during 2000-2015 were estimated by
using a technology-based emission factor method. To support the application of this method,
databases of Hg concentrations in raw materials, technology development trends, and Hg
removal efficiencies of air pollution control devices (APCDs) were constructed through
national sampling and literature review. Hg input to ISP increased from 21.6 t in 2000 to 94.5
t in 2015. In the various types of raw materials, coking coal and iron concentrates contributed
41%-55% and 22%-30% of the total Hg input. Atmospheric Hg emissions from ISP increased
from 11.5 t in 2000 to 32.7 t in 2015 with the peak of 35.6 t in 2013. During the study period,
although sinter/pellet plant and blast furnace were the largest two emission processes,
emissions from roasting plant and coke oven accounted for 22%-34% of ISP's emissions,
which indicated that attention should also be paid on the emissions from these processes when
estimating ISP's emissions. Overall Hg speciation shifted from 50/44/6 (gaseous elemental
Hg ($Hg^0$) / gaseous oxidized Hg ($Hg^{II}$) / particulate-bound Hg ($Hg_p$)) in 2000 to 40/59/1 in
2015, which indicated higher proportion of Hg deposition around the emission points. In the
coming years, emissions from ISP are expected to decrease due to the projection of decreasing
steel productions, increasing energy consumption efficiency, and improvement of APCDs.
With the coming of high-yield-period of steel scrap production, the increasing application
proportion of short process steel making method will not only reduce Hg emissions, but also
increase the emission proportion of $Hg^0$.




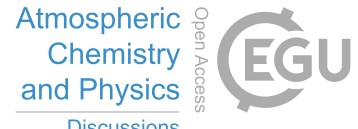

## 1 Introduction

China is the largest iron and steel production (ISP) country in the world. Crude steel production has increased from 127 Mt in 2000 to 804 Mt in 2015 (CISIA, 2001-2016). Rapid growth of ISP has led to large emissions of air pollutants including mercury (Hg) (Wang K. et al., 2016). To reduce Hg pollution, it is important to quantify atmospheric Hg emissions from ISP.

According to existing national inventories, atmospheric Hg emissions from ISP increased from 4.9 t in 1999 to 25.5 t in 2010 (AMAP/UNEP, 2008; Streets et al., 2005; Wu et al., 2006; Zhang L. et al., 2015). In these studies, Hg emissions were determined as the product of crude steel production and unique emission factor of 0.04 g/t steel produced. Later long-term emission inventories revised the unique emission factor with dynamic factors by adopting transformed normal distribution function (Tian et al., 2015; Wang K. et al., 2016; Wu et al., 2016). Such method was based on the assumption that the emission factor was gradually improved according to the simulation curve and attempted to simulate the impact of technology improvement and pollution control on emission factor variation. However, the emission factors actually did not link with technology and APCDs directly. Thus, the simulated emission factors may be quite different from actual situation during a certain short-term period (e.g. ten years) when technology and APCDs experienced dramatic change (Wu et al., 2016), especially under the background of tightening requirement of environmental protection in China in the past decades (MEP, 2011; NEA, 2014; SC, 2013). Recent global assessment report applied a technology-based emission factor method for global ISP including China's (AMAP/UNEP, 2013). However, most of the parameters which were used to calculate the emission factors were from developed countries, which may impact the accuracy of emissions from developing countries such as China. In addition, emissions due to the use of steel scrap were not calculated in the report. With the coming of high-yield-period (after 2020) of steel scrap production in China (Guo and Wei, 2010), emissions due to the consumptions of steel scrap cannot be ignored.

The dominant parameters of a technology-based emission factor included Hg removal efficiencies of APCDs and Hg concentrations in raw materials (Wu et al., 2016; Wu et al.,





2012; Zhang L. et al., 2015). As to Hg removal efficiencies of APCDs, we hypothesized that
the use of data from recent filed experiments on atmospheric Hg emission characteristics in
China's ISP will provide a foundation for the technology-based emission factor model (Wu et
al., 2016; Wu et al., 2012; Zhang L. et al., 2015). However, current studies cannot support the
construction of Hg concentration databases for raw materials. Various raw materials were
used in ISP, covering iron concentrates, iron block, alloy materials, steel scrap, coal, and
additives (mainly limestone and dolomite). Field experiments in three China's steel smelters
indicated that the concentration of iron concentrates was in the range of 23-66 ng/g (Wang F.
Y. et al., 2016a; Zhang L. et al., 2015). However, the Hg concentration data from limited
samples may lead to large uncertainty of the national inventory. Large studies have reported
Hg concentrations in coal (Swaine, 1992; Tian et al., 2010; USGS, 2004; Zhang et al., 2012).
But the specific requirement of low-sulfide coal (less than 1.2%) may make the Hg
concentration in the consumed coal in ISP different from previous databases (Tao and Wang,
1994), since low-sulfide coal was generally companied by low-Hg (Zhang, 2012). Rarely
studies have reported Hg concentrations in steel scrap and dolomite. Therefore, constructing
Hg concentration databases of raw materials was the base to apply a technology-based
emission factor model for China's ISP.
In this study, a technology-based emission factor model was constructed to estimate
atmospheric Hg emissions from China's ISP. To fulfill this aim, raw materials consumed in
steel smelters have been sampled and Hg concentrations have been analyzed to construct the
Hg concentration databases. Up-to-date Hg removal efficiencies from field experiments and
the development trends of production technology and APCDs have been summarized to
support the application of emission factor model.
**2   Methodology**
**2.1  Technology-based emission factor model for ISP**
Generally speaking, ISP method included long process steel making method and short
process steel making method. The long process steel making method included roasting plant,
coke oven, sinter/pellet plant, blast furnace, and oxygen steel making (**Fig. 1**). The short



process steel making method produced crude steel in arc steel making process directly.

Thus, atmospheric Hg emissions from ISP by province can be calculated as follows.

$$
\begin{aligned}
E_i(t) &= E_{i,l}(t) + E_{i,s}(t) \\
&= E_{i,l,r}(t) + E_{i,l,c}(t) + E_{i,l,p}(t) + E_{i,l,b}(t) + E_{i,l,o}(t) \\
&\quad + E_{i,s,a}(t)
\end{aligned}
\tag{E1}
$$

where, $E$ was atmospheric Hg emissions from ISP, t; $i$ was province; $t$ referred to studied

year; $l$ and $s$ referred to long and short process steel making method; $r$, $c$, $p$, $b$, $o$, $a$ referred to
roasting plant, coke oven, sinter/pellet plant, blast furnace, oxygen steel making, and arc steel
making.

For each process $x$, the technology-based emission factor and speciated Hg emissions

can be calculated as follows.

$$
EF_{i,x,k}(t) = \sum_j C_{i,x,j} \times M_{i,x,j}(t) \times S_i(t)^{-1} \times \gamma_x \times \sum_m \theta_m(t) \times \delta_{k,m} \times (1 - \eta_m) \times 1000^{-1}
\tag{E2}
$$

$$
E_{i,x,k}(t) = EF_{i,x,k}(t) \times S_i(t)
\tag{E3}
$$

where, $EF$ was emission factor, g/t; $x$ was studied process; $k$ was speciated Hg; $j$ was the

type of consumed raw material; $C$ was Hg concentration in the consumed raw material, ng/g
(see section 2.2.1); $M$ was the consumption of raw material, Mt (see section 2.2.2); $S$ was the
production of crude steel, Mt (see section 2.2.2). $\gamma$ was Hg release rate, which meant the
percentage of Hg released to flue gas from raw material, %. Hg release rate were collected
from filed experiment studies (**Table S1**). That was 98% for roasting plant, 80% for coke
oven, 85% for sinter/pellet plant, 98% for blast furnace, 80% for oxygen steel making furnace,
and 95% for arc steel making furnace. $m$ referred to the type of APCD combination (see
section 2.2.3); $\delta$ was the proportion of different Hg speciation (see section 2.2.3), %; $\theta$ was
the application rate of different APCD combinations (see section 2.2.3), %; $\eta$ was Hg removal
efficiency (see section 2.2.3), %.
**2.2  Parameters for model**
2.2.1 Hg concentrations in raw materials
For the long process steel making method, the dominant raw materials included iron





concentrates, iron block, coal, limestone, dolomite, alloy, and steel scrap (**Fig. 1**). In the
roasting plant, limestone and dolomite were roasted together or separately to make quick lime
and caustic dolomite. In the coke oven, washed coal was used to produce coke. In the
sinter/pellet plant, iron-containing materials (mainly iron concentrates), quick lime, caustic
dolomite, and produced coke were mixed to produce sintered/pellet block, which were used as
raw materials with coal and produced coke in the blast furnace. The produced pig iron from
blast furnace and additional scraps were used to produce steel in the oxygen steel making
processes. For the short process steel making method, arc steel making process was applied to
produce steel by mainly using scraps as raw materials. In each process, Hg input due to the
use of intermediated products (e.g., quick lime, caustic dolomite, and coke) was calculated by
using mass balance method (Wu et al., 2012).

National sampling and Hg concentration analysis were conducted to construct the Hg

concentration databases for the consumed raw materials. The sampling, preparation and
analysis methods were described in detail in our previous studies (Wu et al., 2012; Zhang et
al., 2012). Lumex 915M + pyro attachment (with a detection limit of 0.5 ng/g) was applied to
analyze Hg concentration by using U.S. EPA Method 7473 (US EPA, 1998). Number of
samples and Hg concentrations in dominant raw materials by province were shown in **Table 1**.
National Hg concentrations (median value) in the consumed iron ore were 20 (0.6-387) ng/g,
which were lower than the median value of 30 (0.6-600) ng/g used in the global assessment
report (AMAP/UNEP, 2013). Overall Hg concentrations in the consumed coking coal (82
ng/g) and pulverized coal injection (PCI) coal (73 ng/g) were lower than the 170 (8-2248)
ng/g (Zhang, 2012) used in China's coal combustion sectors but higher than the 55 (50-60)
ng/g of global assessment report. Hg concentrations in the limestone were 18 (0.9-2753) ng/g.
Although the median value was lower than value of 30 (20-50) ng/g applied in the global
assessment report, the variation range was much wider according to our analysis. Hg
concentrations (median value) in the dolomite and iron block were 9 ng/g and 19 ng/g. In
oxygen and arc steel making processes, the main iron-containing materials were steel scrap,
alloy scrap, and pig iron. Hg concentrations (median value) in steel scrap and alloy scrap were
48 and 2 ng/g while the concentrations in pig iron were less than detection limit. For province




with samples no less than 15, the distribution characteristics of Hg concentrations of the
samples were generated by using the batch fit function of Crystalball software. Otherwise, Hg
concentrations were assumed to fit normal distribution.
2.2.2 Provincial consumptions of raw materials

Provincial consumptions of raw materials in 2015 were shown in **Table S2**. National

limestone consumptions were converted from quick lime consumptions (Ma, 2011) by using
the factor of 1.95 t limestone to produce 1 t quick lime (CISIA, 2001-2016) (**Table S3**).
National dolomite consumptions were derived from China steel statistics report (Ma, 1995)
according to the production trends of crude steel. The limestone and dolomite can be
consumed in the roasting and sinter/pellet plant. In the sinter/pellet plants, additives
(including limestone and dolomite) consumptions were approximately 153.9 kg/t sinter
produced or 10.5 kg/t pellet produced (CISIA, 2001-2016). We assumed that 88% of the
additives were limestone and 12% were dolomite according to filed experiments (Wang F. Y.
et al., 2016). The rest of limestone and dolomite were consumed in the roasting plants.
Provincial consumptions of limestone and dolomite were distributed according to the
proportions of provincial pig iron productions in national production (Table S2). Provincial
pig iron productions were collected directly from yearbooks (CISIA, 2001-2016).

Provincial coking coal consumptions were converted from provincial coke consumptions

(CISIA, 2001-2016). Generally, there were two main types of coke production methods,
including machining coke production method and indigenous coke production method. Coal
consumptions were 1.35 t to produce 1 t of machining coke or 1.65 t to produce 1 t of
indigenous coke (UNEP, 2013; Wang, 1991). The produced cokes were used as raw materials
in both sinter/pellet plant and blast furnace. Provincial coke consumptions in blast furnace
were converted according coke ratio of 363-388 kg coke per t pig iron produced (CISIA,
2001-2016). The rest of cokes were assumed to be consumed in sinter/pellet plant.

National iron concentrate consumptions were converted from sinter/pellet productions.

Approximately 0.91-0.92 t and 0.96-0.97 t of iron concentrates were needed to produce 1 t
sinter and 1 t pellet, respectively (CISIA, 2001-2016). National sinter and pellet productions
were obtained directly from yearbooks (CISIA, 2001-2016) and provincial data were





converted according to provincial pig iron productions.
National PCI coal consumptions in blast furnace were collected from national energy
statistical yearbook (NESA, 2001-2016) and the provincial data were converted according to
provincial pig iron productions. The iron block consumption in blast furnace were converted
from pig iron production by using the factor of 156 kg iron block per t pig iron produced
(CISIA, 2001-2016).
The steel scrap was consumed in both oxygen and arc steel making process.
Consumptions of steel scrap in oxygen and arc steel making process were approximately 59.4
and 361.9 kg/t crude steel produced. Alloy consumptions to produce per t crude steel were
16-17 kg in oxygen steel making process and 140-156 kg in arc oxygen making process.
Oxygen and arc steel productions were collected directly from yearbooks (CISIA, 2001-2016).
Based on these ratios, provincial steel scrap and alloy consumptions were converted from
provincial crude steel productions.
2.2.3 Application rate, Hg removal efficiency, and Hg speciation
In the roasting plant, blast furnace, and steel making process, dust collectors such as
venturi, cyclone (CYC), wet scrubber (WS), electrostatic precipitator (ESP), and fabric filter
(FF) were used for flue gas dedusting. In coke oven process, the coal was firstly washed
before consumed and the flue gas was cleaned with dust collectors or with additional washing
scrubbers. For flue gas from sinter/pellet plant, they were generally cleaned with dust
collectors. Additional flue gas desulfurization towers (FGD) were gradually applied after
2010. The application rates of different APCD combinations by process during 2005-2010
were collected from previous studies (Wang et al., 2014; Zhao et al., 2013). The data of
2000-2004 and 2011-2015 were mainly derived from yearbooks (CISIA, 2001-2016; NBS,
2001-2016) (Table S4). Hg removal efficiencies and speciation profiles of APCDs (**Table S4**)
were collected from filed experiments and literature on emission studies (Gao, 2016; Wang F.
Y. et al., 2016; Zhang L. et al., 2015). The distribution characteristics of Hg removal
efficiencies were assumed to fit normal distribution characteristics.





### 2.3 Uncertainty analysis


Monte Carlo simulation was introduced to estimate the uncertainty of emissions.
Detailed description of the simulation processes has been reported in our previous studies
(Hui et al., 2016; Wu et al., 2016; Zhang L. et al., 2015). In this study, the (P50-P10)/P50 and
(P90-P50)/P50 values were still regarded as lower and upper limits of uncertainties with 80%
confidence degree, where P10, P50, and P90 meant that the probabilities of actual results
lower than corresponding values were 10%, 50%, and 90%, respectively.

### 3     Results and Discussion


### 3.1 Hg input trends


Hg input to ISP increased from 21.6 t in 2000 to 94.5 t in 2015 (**Fig. 2**). The peak of Hg
input was in 2014 when the crude steel production reached the highest value (**Table S1**).
During 2000-2014, the average annual growth rate (AAGR) of Hg input was 11% while Hg
input reduced by 3% from 2014 to 2015. In the various types of raw materials, coking coal
and iron concentrates contributed the largest amount of Hg input, accounting for 35%-46%
and 25%-32% of the total, respectively. Hg input due to the use of coking coal increased from
9.9 t in 2000 to 33.5 t in 2015. Hg input with iron concentrates increased at AAGR of 12%
from 2000 and reached 29.2 t in 2015. The PCI coal brought approximately 6%-9% of Hg to
ISP. Hg in the additives (including limestone and dolomite) contributed 12%-18% of total Hg
input. Hg in the iron block was in the range of 0.6-3.6 t. Hg input due to the use of steel scrap
and alloy was 4.1 t in 2015, accounting for 4% of national total. However, steel scrap and
alloy contributed 7% of total crude steel production in 2015 (CISIA, 2016).

### 3.2 Hg emission trends


3.2.1 Hg emission trends by process
Atmospheric Hg emissions from ISP increased from 11.5 t in 2000 to 32.7 t in 2015 (**Fig.**
**3**). The peak of emissions was in 2013 when the emissions reached 35.6 t. In 2015, emissions
from long process steel making method and short process steel making were 32.2 t and 0.5 t,
accounting for 98.3% and 1.7% of national total, respectively. Thus, emissions from long
process steel making were still the dominant source of Hg emissions from China's ISP.



Among the processes, emissions from sinter/pellet plant accounted for 42%-49% of annual
total. Its emissions increased from 4.8 t in 2000 at the AAGR of 10.1% and reached 15.9 t in
2015. Blast furnace was also significant Hg emission process. Its emissions increased from
1.9 t in 2000 to 7.9 t in 2015 at AAGR of 10.0%. AAGR for roasting plant and coke oven was
8.3% and 1.2%. In 2015, both emissions from roasting plant and coke oven were 3.5 t,
respectively.
The slower AAGR of Hg emissions (7.2%) than that of crude steel production (13%)
reflected the impact on Hg emission reduction due to energy saving and environmental
protection in ISP. On one hand, Hg input to produce unitary crude steel decreased from 0.17
to 0.12 g/t, which mainly benefited from the improvement of coke production efficiency and
energy utilization efficiency of sinter/pellet plant and blast furnace. Since 2004, indigenous
coke production method with high coal consumption has been gradually replaced with
machine coke production method. The coke ratio in sinter/pellet plant has been reduced from
approximately 388 kg/t pig iron produced in 2000 to 363 kg/t in 2015 (CISIA, 2001-2016).
On the other hand, the improvement of APCDs increased the overall Hg removal efficiency
from 54% in 2000 to 72% in 2015. APCDs for coke oven have shown the largest Hg removal
efficiencies (64%-87%) while the improvement of APCDs in sinter/pellet plant contributed
most to the rapid Hg reduction speed during 2000-2015. The replacement of CYC and WS
with ESP and FF in sinter/pellet plant improved Hg removal efficiency from 21% in 2000 to
44% in 2010. The application of FGD in addition to dust collectors was the main driver of Hg
reduction in sinter/pellet process during 2011-2015. Hg removal efficiency in sinter/pellet
plant was 53% in 2015.
3.2.2 Hg emission trends by province
Provincial Hg emissions in 2000, 2005, 2010, and 2015 were shown in **Table 2**. In 2000,
Shanxi, Shanghai, Henan, Hebei, and Shandong were the top five largest emitters with
emissions larger than 1 t. Emissions from these five provinces contributed to 58% of national
Hg emissions. Following these five provinces were Liaoning, Beijing, Gansu, Jiangsu, and
Jiangxi. Summation of the emissions from all the above ten provinces were 9.0 t, accounting
for 78% of national emissions in 2000. At the provincial level, we noted significant



differences of Hg emission trends during the past 16 years. The AAGR of provincial Hg
emissions varied from -40% to 26%. Negative AAGR existed in Beijing and Shanghai
provinces, the two most economically developed regions in China. Hg abatement in these two
regions was mainly caused by the reduction of crude steel production, which was transferred
to nearby provinces such as Hebei, Zhejiang, Jiangsu, and Shandong. Thus, Hg emissions in
these nearby provinces all presented high AAGR of more than 10%. In 2015, the largest five
Hg emission provinces were changed to Hebei, Shandong, Henan, Jiangsu, and Shanxi
provinces. Emissions from these provinces reached 21.5 t, accounting for 68% of national
emissions. Liaoning, Jiangxi, Inner Mongolia, Gansu, and Shanghai were also in the list of the
top ten largest emitters.
3.2.3 Hg emission trends by species

Overall Hg speciation profile of ISP experienced great change during the study period,

from 50/44/6 (gaseous elemental Hg ($Hg^0$)/gaseous oxidized Hg ($Hg^{II}$)/particulate-bound Hg
($Hg_p$)) in 2000 to 40/59/1 in 2015 (**Fig. 4**). The proportion of $Hg^{II}$ increased 15%, whereas
both $Hg^0$ and $Hg_p$ proportion showed decreasing trend. Such shift indicated higher deposition
proportion of Hg around the emission points since $Hg^{II}$ has larger deposition velocity and
higher water-solubility. For the long process steel making, Hg speciation profile shifted from
49/44/7 in 2000 to 39/60/1. The speciation shift in roasting plants was mainly impacted by the
replacement of WS and CYC with FF, which increased the emitted $Hg^{II}$ proportion from 38%
to 75%. The replacement of indigenous coke production method with machine coke
production method mainly contributed to $Hg^{II}$ proportion increase from 42% to 52% at first.
However, the gradual installation of WS in addition to cooler for air pollution control of
machine coke production method further washed $Hg^{II}$ and reduced $Hg^{II}$ proportion to 49% in
2015. $Hg^{II}$ proportion in the exhaust gas of sinter/pellet plant has increased by 20%. The
increase of $Hg^{II}$ proportion in sinter/pellet plant was mainly impacted by the substitute of WS
with ESP, FF, ESP+WFGD, or ESP+DFGD+FF which generally emitted gas with higher $Hg^{II}$
proportion (**Table S4**). Increase of $Hg^{II}$ emission proportion in blast furnace was due to higher
$Hg^{II}$ emission proportion after FF than venturi. In the oxygen steel making process, Hg
speciation profile almost unchanged. For the short process steel making, $Hg^0$ was the



dominant speciation during the whole study period and the proportion of $Hg^0$ increased from
66% in 2000 to 79% in 2015.

**3.3   Uncertainty analysis**

In 2015, overall uncertainty of atmospheric Hg emissions from ISP was in the range of
(-29%, 77%) (**Fig. 5**). Previous studies in ISP indicated the emission uncertainties of this
source were (-80%.100%) in the study of Zhang et al. (2015) and (-100%, 100%) in the study
of Streets et al. (2005) and Wu et al. (2006). The improvement of emission estimation of this
study was contributed by better knowledge on the Hg concentrations of raw materials and Hg
removal efficiencies of APCDs. In all ISP processes, the largest uncertainties existed in
emissions from roasting plant (-59%, 130%) and sinter/pellet process (-45%, 126%). These
mainly due to larger distribution range of Hg concentrations in limestone and iron ore as well
as Hg removal efficiencies of APCDs. The uncertainties of Hg emissions from other processes
were much lower, (-49%, 48%) for coke oven, (-23%, 46%) for blast furnace, (-41%, 27%)
for oxygen steel making, and (-60%, 54%) for arc steel making.

**3.4   Comparison and implications**

Due to the complicated ISP processes and limitations of data availability, the process
combinations included in different emission inventories was divided in to four types (**Fig. 6**)
(AMAP/UNEP, 2008, 2013; Wang K. et al., 2016; Wu et al., 2016; Wu et al., 2006; Zhang L.
et al., 2015). The first type included sinter plant and blast furnace, which were the basic
assumption in the emission inventories of Wu et al. (2003), Zhang L. et al. (2015), and
AMAP/UNEP (2005). In these studies, unique emission factor of 0.0400 g/t was applied
(**Table S5**) and their emissions were similar in the same inventory year. Our emissions for this
process combination were almost the same as above estimations around 2005. However, the
gap grew with time when FGD was gradually applied in sinter/pellet plant. Therefore,
emission factor for this type of combination was reduced from 0.0527 g/t in 2000 to 0.0296
g/t in 2015 (**Table S5**). The second type also consisted of steel making in addition to the first
type. Our estimation was much higher than the study of Wang K. et al. (2016) because the
emission factors applied in Wang K.'s study were mainly derived from European technical



report (EMEP/CORINAIR, 2001; EMEP/EEA, 2013). However, the technology applied in
Europe may be better than China's situation. For example, the emission factor of 0.00019 g/t
applied for blast furnace with FF was used as best emission factor in Wang K.'s study (2016).
However, the combination of WS and venturi scrubber was the dominant APCD type in
China's blast furnace (Zhao et al., 2013), Hg removal efficiency of which was lower than that
of FF. The third type of process combination also included coke oven as part of ISP in
addition to the second type. Lower Hg emissions estimated by AMAP/UNEP (2013) were due
to their lower Hg concentration in coal. In addition, although different processes were
considered in the report, unique APCDs profile was applied in different process where the
application of ESP+FGD reached 55%. However, FGD were mainly installed in sinter/pellet
plant but rarely applied in other processes. The forth type also considered the emissions from
roasting process where emissions accounted for 9%-11% of total emissions.
The comparison of emissions from different types of process combination in this study
indicated the significance of our new inventory. The proportion of emissions from these two
processes accounted for 22%-34% of ISP's emissions during the whole study period, which
indicated the importance of including emissions from roasting plant and coke oven in the ISP
emission inventories. In addition, given the impact of APCDs on the emission estimation,
inventories in ISP should also apply distrinct APCDs profiles for different processes so as to
reduce the uncertainty of inventories. In the inter-annual emission inventories, the
technology-based estimation method by linking emissions with technology and APCDs
directly has shown its advantage in the discussion of emission trends and quantification of Hg
removal due to air pollutants control measures.
Based on the comprehensive consideration of dominant parameters (e.g., steel
production, air pollution control measures) in the technology-based method and the emission
trends from China's ISP during 2000-2015, we expected a decreasing trend of Hg emissions
in the coming years. On one aspect, emissions of pollutants are required to be reduced by 15%
for ISP before 2020 in China (MIIT, 2016). To fulfill this goal, corresponding emission
standards have been issued (Wu, 2013), which will accelerate the applications of improved
APCDs. During 2010-2015, the increase of $SO_2$ emission limits from 1500 mg/m$^3$ to 200





mg/m³ promoted the large-scale application of desulfurization devices in the sinter/pellet plant
of ISP. After 2015, ISP will move forward to $NO_x$ control by using related technologies such
as selective catalytic reduction (SC, 2013), the synergic Hg removal efficiencies of which
have been proved in other industries (Wang et al., 2010). On the other aspect, excess steel
production capacity combined with decreasing steel consumption currently will lead to the
reduction of steel production, as what we have seen during 2014-2015 (CISIA, 2016; MIIT,
2016). The crude steel production trends will thereby substantially reduce Hg input to ISP.
Besides, energy consumptions are required to be reduced by more than 10% during the
2016-2020 (MIIT, 2016). The improvement of energy efficiency in the main processes will
also be one dominant measure to reduce energy consumption (Li et al., 2015), which will
reduce fuel consumption and further lead to the reduction of Hg input. In addition, with the
coming of high-yield-period of steel scrap production (Guo and Wei, 2010), the application
proportion of short process steel making method is expected to increase according to the
experiences in developed countries (e.g., Europe), which will indirectly reduce the
requirement of steel production from long process steel making method. The replacement of
steel production method will also be one driver of Hg emission reduction considering lower
Hg emission factors of short process. In addition, since $Hg^0$ is the dominant Hg speciation
from arc steel making process, the emission proportion of $Hg^0$ is expected to increase.

### 4    Conclusion

In this study, updated atmospheric Hg emissions from ISP during 2000-2015 were
estimated by using technology-based emission factor method with up-to-date parameters. The
input of Hg as impurity of raw materials for ISP increased from 21.6 t in 2000 to 94.5 t in
2015. In the various types of raw materials, coking coal and iron concentrates contributed to
the largest amount of Hg input, 35%-46% and 25%-31% of national total, respectively.
Atmospheric Hg emissions from ISP increased from 11.5 t in 2000 to the peak of 35.6 t in
2013, and then reduced to 32.7 t in 2015. Overall Hg speciation shift from 50/44/6
($Hg^0/Hg^{II}/Hg_P$) in 2000 to 40/59/1 in 2015. In the coming years, emissions from ISP are
expected to decrease due to the projection of Hg input reduction and improvement of APCDs.
In 2015, emissions from long process steel making method and short process steel





making were 32.2 t and 0.5 t, accounting for 98.3% and 1.7% of national total, respectively.
Sinter/pellet plant and blast furnace were the largest two emission processes, emissions from
which accounted for 49% and 24% of national emissions. However, emissions from roasting
and coke oven should cause attention because their emissions accounted for 22% of national
emissions. The largest five Hg emission provinces were Hebei, Shandong, Henan, Jiangsu,
and Shanxi provinces. Emissions from these provinces reached 21.5 t, accounting for 68% of
national emissions.

With better understanding of Hg flow in ISP, the uncertainty of atmospheric Hg

emissions from ISP estimated by using technology-based emission factor model has largely
reduced. However, with the continuously change of APCD combinations, extensive and
dedicated field experiments are still required to generate suitable database of Hg removal
efficiencies for the improved APCDs in the future.
*Acknowledgment.* This study was supported by the Major State Basic Research Development
Program of China (973 Program) (2013CB430001), Natural Science Foundation of China
(21607090), and China Postdoctoral Science Foundation (2016T90103, 2016M601053)



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


**Tables**
Table 1. Hg concentration in the raw materials



| Province[1] | Iron ore AM[2] | Iron ore MV[2] | Iron ore SV[2] | Iron ore NS[2] | Limestone AM | Limestone MV | Limestone SV | Limestone NS | Dolomite AM | Dolomite MV | Dolomite SV | Dolomite NS | Coking coal AM | Coking coal MV | Coking coal SV | Coking coal NS | PCI coal AM | PCI coal MV | PCI coal SV | PCI coal NS |
|---|---|---|---|---|---|---|---|---|---|---|---|---|---|---|---|---|---|---|---|---|
| | ng/g | ng/g | ng/g | | ng/g | ng/g | ng/g | | ng/g | ng/g | ng/g | | ng/g | ng/g | ng/g | | ng/g | ng/g | ng/g | |
| Tianjin | 41 | 44 | 8 | 3 | 9 | 9 | 3 | 1 | 5 | 5 | 2 | 4 | | | | | 75 | 66 | 31 | 5 |
| Hebei | 37 | 24 | 40 | 79 | 374 | 117 | 631 | 27 | 7 | 8 | 5 | 5 | 78 | 66 | 39 | 71 | 79 | 85 | 70 | 38 |
| Shanxi | 30 | 24 | 28 | 25 | 9 | 7 | 8 | 6 | | | | | 77 | 71 | 27 | 22 | 114 | 106 | 19 | 3 |
| Shanghai | 44 | 44 | 50 | 4 | 20 | 20 | 5 | 1 | 39 | 39 | 2 | 2 | 125 | 152 | 71 | 5 | 12 | 12 | 0 | 2 |
| Jiangsu | | | | | 103 | 68 | 113 | 18 | | | | | | | | | | | | |
| Zhejiang | | | | | 52 | 37 | 35 | 22 | | | | | | | | | | | | |
| Anhui | 21 | 20 | 12 | 14 | 11 | 11 | 4 | 12 | 32 | 32 | 2 | 3 | 88 | 58 | 53 | 8 | 98 | 22 | 104 | 7 |
| Fujian | 19 | 14 | 11 | 15 | 11 | 11 | 5 | 4 | 9 | 8 | 5 | 4 | 105 | 100 | 51 | 13 | 255 | 239 | 51 | 7 |
| Jiangxi | 57 | 46 | 88 | 18 | | | | | 10 | 9 | 4 | 3 | 120 | 93 | 59 | 4 | 185 | 198 | 66 | 7 |
| Shandong | 125 | 128 | 73 | 20 | | | | | 24 | 24 | 1 | 2 | 72 | 69 | 21 | 12 | | | | |
| Henan | 32 | 24 | 29 | 18 | 692 | 759 | 629 | 13 | | | | | | | | | 131 | 116 | 91 | 21 |
| Hubei | | | | | 16 | 14 | 8 | 6 | | | | | 92 | 83 | 54 | 39 | | | | |
| Hunan | 33 | 19 | 34 | 77 | 102 | 87 | 62 | 4 | 3 | 3 | 1 | 2 | | | | | | | | |
| Guangdong | | | | | 48 | 44 | 26 | 15 | | | | | | | | | | | | |
| Guangxi | | | | | 8 | 8 | 2 | 4 | | | | | 210 | 188 | 112 | 27 | 96 | 98 | 30 | 6 |
| Chongqing | | | | | | | | | | | | | 85 | 79 | 48 | 20 | 65 | 65 | 12 | 4 |
| Sichuan | 37 | 2 | 44 | 7 | 10 | 9 | 4 | 12 | | | | | | | | | | | | |
| Guizhou | | | | | 11 | 10 | 11 | 42 | | | | | | | | | | | | |
| Yunnan | 10 | 10 | 7 | 6 | 17 | 20 | 7 | 6 | | | | | 51 | 52 | 7 | 8 | 58 | 56 | 6 | 3 |





| | | | | | | | | | | | | | | | | | | | | | |
|---|---|---|---|---|---|---|---|---|---|---|---|---|---|---|---|---|---|---|---|---|---|
| Gansu | 107 | 107 | 3 | 3 | 3 | 1 | 1 | | | | 166 | 174 | 21 | 5 | 60 | 60 | 5 | 2 | | | |
| Xinjiang | 6 | 5 | 4 | 3 | 17 | | | | | | 82 | 82 | 46 | 22 | 31 | 24 | 19 | 15 | | | |
| National | 38 | 20 | 48 | 306 | 153 | 18 | 402 | 204 | 14 | 9 | 12 | 25 | 99 | 82 | 68 | 256 | 99 | 73 | 83 | 120 | | |

Note: 1. Provinces without data were not listed in this table;
2. AM: Average mean; MV: Median value; SV: Standard value; NM: Number of samples.



Table 2. Provincial Hg emissions during 2000-2015

| Province | Atmospheric Hg emissions (t) | | | | |
|---|---|---|---|---|---|
| | 2000 | 2005 | 2010 | 2015 | AAGR |
| Beijing | 0.4 | 0.3 | 0.1 | 0.0 | -40% |
| Tianjin | 0.3 | 0.5 | 0.9 | 0.7 | 7% |
| Hebei | 1.2 | 4.0 | 6.7 | 7.1 | 13% |
| Shanxi | 1.6 | 2.2 | 2.1 | 1.8 | 1% |
| Inner Mongolia | 0.3 | 0.5 | 0.7 | 0.8 | 7% |
| Liaoning | 0.8 | 1.2 | 1.7 | 1.6 | 4% |
| Jilin | 0.1 | 0.2 | 0.3 | 0.3 | 6% |
| Heilongjiang | 0.1 | 0.2 | 0.3 | 0.2 | 6% |
| Shanghai | 1.5 | 1.2 | 1.0 | 0.7 | -5% |
| Jiangsu | 0.3 | 1.1 | 1.9 | 2.5 | 15% |
| Zhejiang | 0.1 | 0.1 | 0.4 | 0.4 | 11% |
| Anhui | 0.3 | 0.4 | 0.6 | 0.6 | 6% |
| Fujian | 0.1 | 0.2 | 0.2 | 0.3 | 11% |
| Jiangxi | 0.3 | 0.7 | 1.1 | 1.0 | 8% |
| Shandong | 1.1 | 4.0 | 6.0 | 6.1 | 12% |
| Henan | 1.3 | 2.2 | 3.5 | 4.0 | 8% |
| Hubei | 0.3 | 0.4 | 0.5 | 0.4 | 2% |
| Hunan | 0.2 | 0.5 | 0.7 | 0.6 | 7% |
| Guangdong | 0.1 | 0.3 | 0.3 | 0.4 | 8% |
| Guangxi | 0.1 | 0.2 | 0.4 | 0.5 | 13% |
| Hainan | 0.0 | 0.0 | 0.0 | 0.0 | 26% |
| Chongqing | 0.1 | 0.1 | 0.2 | 0.1 | 2% |
| Sichuan | 0.2 | 0.4 | 0.4 | 0.4 | 4% |
| Guizhou | 0.1 | 0.2 | 0.2 | 0.2 | 4% |
| Yunnan | 0.1 | 0.3 | 0.4 | 0.3 | 7% |
| Tibet | 0.0 | 0.0 | 0.0 | 0.0 | / |
| Shaanxi | 0.1 | 0.2 | 0.4 | 0.7 | 16% |
| Gansu | 0.4 | 0.5 | 0.6 | 0.7 | 4% |
| Qinghai | 0.0 | 0.0 | 0.1 | 0.0 | 5% |
| Ningxia | 0.0 | 0.0 | 0.1 | 0.1 | 19% |
| Xinjiang | 0.1 | 0.1 | 0.3 | 0.3 | 13% |
| Total | 11.5 | 22.2 | 31.9 | 32.7 | 7% |






**Figures**

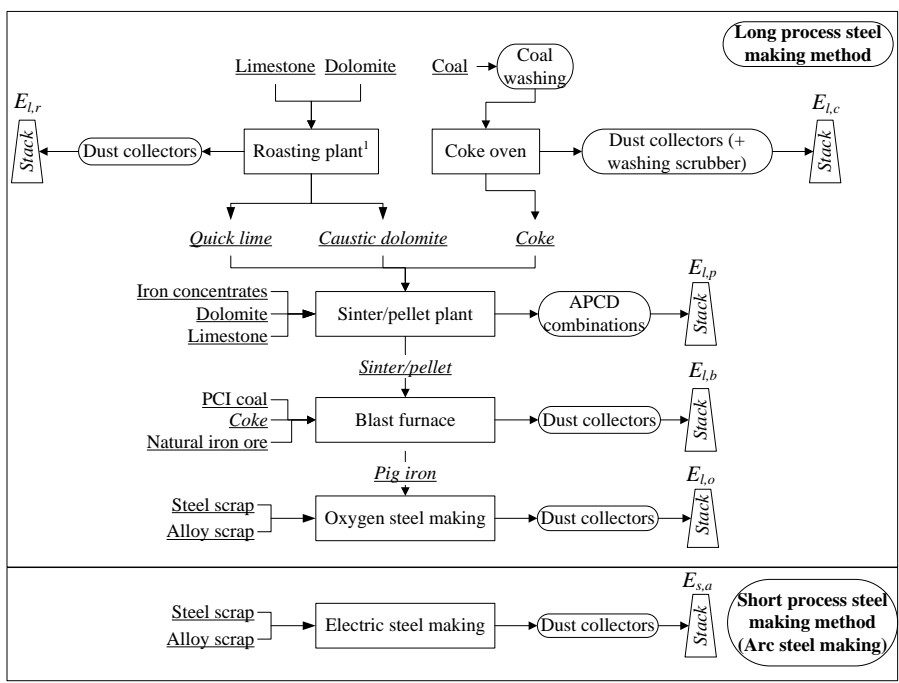


**Fig. 1**. Flow chart of ISP processes (1. Some plants roasted limestone and dolomite separately.)

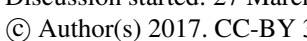



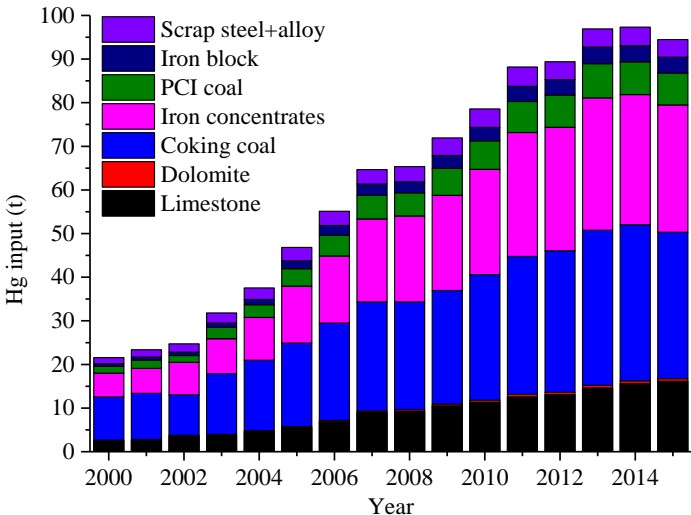


**Fig. 2**. Hg input trends by material




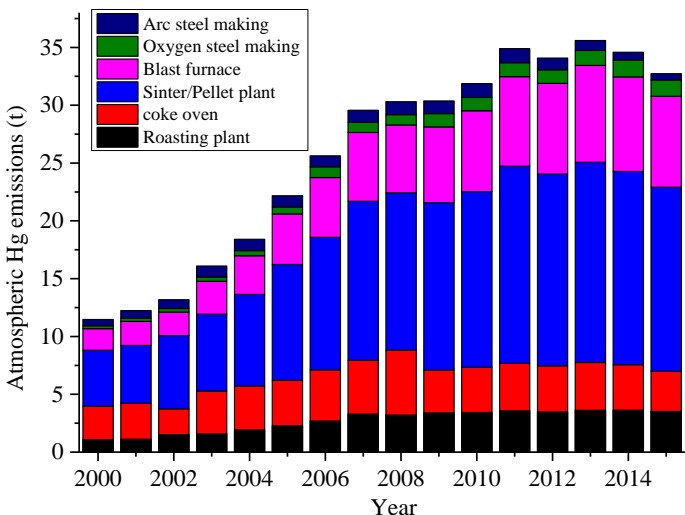


**Fig. 3**. Hg emission trends by process





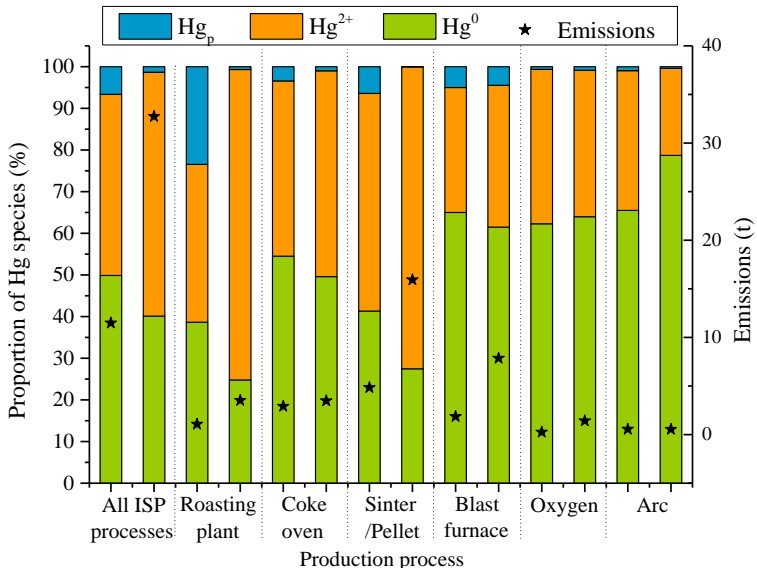


**Fig. 4**. Proportion of different Hg species (For each process, the left and right column
represents the data in 2000 and 2015, respectively)





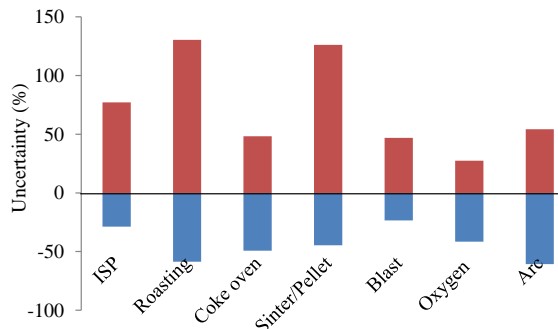


**Fig. 5**. Unertainty analysis





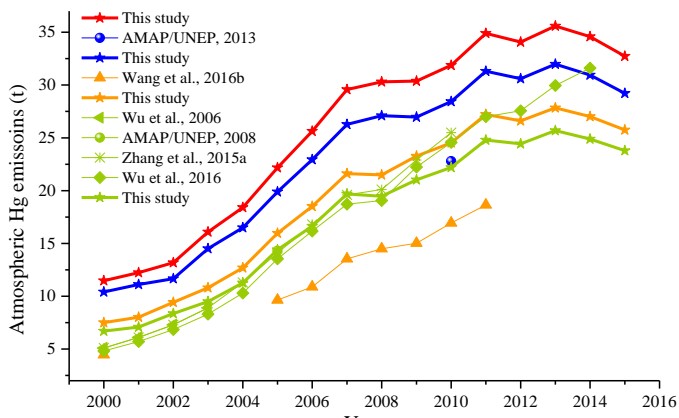

Red symbol: Sinter plant+blast furnace+roasting plant+coke oven+steel making
Blue symbol: Sinter plant+blast furnace+coke oven+steel making
Orange symbol: Sinter plant+blast furnace+steel making
Green symbol: Sinter plant+blast furnace


**Fig. 6**. Atmospheric Hg emissions of ISP in different studies