# Peer review of "Updated atmospheric speciated mercury emissions from"

_Atmospheric Chemistry and Physics, 2017_

## Referee Comment (RC1) · Anonymous Referee #1 · 30 Apr 2017

Authors estimated mercury emission from China's iron and steel production (ISP) during 2000-2015 by using a technology-based emission factor method. They presented trends of Hg input to and emission from ISP and further differentiate them into detailed sectors, and noted that Hg emissions from roasting plant and coke oven cannot be overlooked.

The paper is technically good. My primary concern is that the paper is lack of enough innovation. Actually, authors recently presented anthropogenic mercury emission from all sectors including ISP (Wu et al., 2016, ES&T), and in this paper, authors basically follow the ideas and methods to further update mercury emission of ISP in the past 15 yrs. Hg emission of ISP peaked at 35.6 t in 2013 and is not a major contributor to the total Hg emission of 530t in 2014 as estimated in authors' previous work.

[Figure]

Compared with previous studies, this updated ISP inventory included consideration of roasting plant and coke oven, counting for 22%-34% of ISP's emissions. I think authors should include some words in introduction to highlight this consideration and stress that these two processes are potentially important in shaping the trends of ISP Hg emissions.

In Wu et al 2016 ES&T paper, authors noted that the Hg emissions from ISP are quite possible to increase, however, in this updated ISP inventory, authors argued that Hg emission from ISP are expected to decrease. I think authors should explain this point clearly and included some discussion in section 3.4.

It seems that Hg release rates used in this study (Table S1) are simply averaged by all available data, why?

Other specific points:

Line70, many studies.

L102, "filed experiment" should be "field experiment", some other places should be also revised.

L206, Table S3

---

## Referee Comment (RC2) · Anonymous Referee #2 · 13 May 2017

General comments:

The authors made a great effort on the estimation of Hg emission from iron and steel production (ISP) during 2010-2015 in China by using technically-based emission factor method with up-to-date parameters through national sampling and literature review, which has reduced the associated uncertainty of Hg emission from ISP. Given the updated Hg emission database, this work provides meaningful results to health-related study and is helpful to make control policy. Frankly speaking, many sections in this paper just reported the numbers but didn't provide further insight beyond those facts.

Specific comments:

My main concern is the credible of dataset. Does the "SV" in Table 1 means standard deviation value? If it does, how can the authors get SV when they just have one

[Figure]

Limestone sample in Shanghai and Gansu? And in Table 2, the emissions of Hg in Hainan in 2000/2005/2010/2015 were 0.0 but it has an average annual growth rate of 26%. Is that because the emission amount of Hg in Hainan was quite low so that it doesn't show the true value when you only keep one decimal fraction? In addition to this, what is the "All ISP processes" means in Fig.4? If it means the sum of Hg emission from all processes, why the total emission of Hg (asterisk in Fig.4) in Roasting plant, Coke oven, Sinter/Pellet, Blast furnace, Oxygen and Arc far exceeding the value in All ISP processes, either in 2000 or 2015?

Given that the annual Hg emission proportion (by comparing Fig. 2 and Fig. 3) shows a decreasing trend since 2000, I'd suggest authors add the emission proportion values to Figure 3 because it will provide more insight of the impact of energy consumption and air pollution control on Hg emission.

Other specific points:

Line 323: the authors stated ". . . indicated the significance of our new inventory", which makes me confused. Please specify what it exact means.

Line 335: the authors quoted "emissions of pollutants are required to be reduced by 15% for ISP before 2020 in China". That is unclear to me what the pollutants really are? Does the government required to reduce all pollutants by 15% simultaneously?

Technical correction:

Line 62, 102, 152 and some other places: "filed" should be "field".

Table 1: "NM" in Note 2 should be revised as "NS".

Figure 1: what is the numeric "1" in the title of figure 1 used for?

[Figure]

---

## Author Comment (AC1) · 27 Jun 2017

Dear editor,
We have revised our manuscript carefully based upon the check list. We have attached a detailed, point-by-point list of our responses to the reviewer's comments.

**Response to reviewer 1**

Authors estimated mercury emission from China's iron and steel production (ISP) during 2000-2015 by using a technology-based emission factor method. They presented trends of Hg input to and emission from ISP and further differentiate them into detailed sectors, and noted that Hg emissions from roasting plant and coke oven cannot be overlooked.

The paper is technically good. My primary concern is that the paper is lack of enough innovation. Actually, authors recently presented anthropogenic mercury emission from all sectors including ISP (Wu et al., 2016, ES&T), and in this paper, authors basically follow the ideas and methods to further update mercury emission of ISP in the past 15 yrs. Hg emission of ISP peaked at 35.6 t in 2013 and is not a major contributor to the total Hg emission of 530t in 2014 as estimated in authors' previous work.

Response: The method in this study is actually different from that applied for ISP in our previous study (Wu et al., 2016, ES&T). Transformed normal distribution function was used to generate the dynamic emission factors in the study of Wu et al., 2016, ES&T. Such method was based on the assumption that the emission factors were gradually improved according to the simulation curve due to technology improvement and pollution control. Basic parameters for such method is quite easy to collect compared to the technology-based emission factor method. In addition, due to lack of information, this method, which was widely applied in the long-term emission inventories (Streets et al., 2011; Tian et al., 2015), was the best method we can apply to estimate the emissions from ISP in Wu et al. 2016.

However, the emission factors generated from transformed normal distribution function actually did not link with technology and APCDs directly. In the study of Wu et al., 2016, we noted that when technology and APCDs experienced dramatic change, the simulated emission factors actually cannot reflect the impact from technology and APCDs as those emission factors calculated by technology-based emission methods (See Figure S3 of Wu et al., 2016 and the following figure). Moreover, the emission contribution of ISP will increase and be one potential large emission source in the future by using this method for ISP, when emissions from large sectors such as coal-fired power plants and nonferrous metals smelting were efficiently controlled according to the requirements of *Minamata Convention on Mercury*. Because the controlled emission sources in the Annex D of the convention can be revised with the change of sectoral emissions in the future, it is quite possible that ISP will also be included in Annex D. Therefore, it is important to better quantify atmospheric Hg emissions from ISP by using technology-based emission factor method and objectively predict its future emissions.

The innovation of this study included the estimation method and our findings. The technology-based emission factor method was established based on the investigation of the complicated ISP processes and up-to-date parameters, which has reduced the associated uncertainty of Hg emission from ISP. Hg concentrations of the various raw materials were collected based on national sampling, including 306 iron ores, 204 limestones, 25 dolomites,

256 coking coal, and 120 PCI coal. These samples were analyzed by using similar analysis method and will be more comparable. As to the results, we noted that the overall Hg removal efficiency was increased from 47% to 65%, which indicated a continuous technology improvement in China's ISP. Emissions by process indicated that the contribution of roasting plant and coke oven cannot be ignored in the ISP's emission inventories.

[Figure]

GB 4913-85: Emission standards for pollutants from heavy non-ferrous metal industry
GB 9078-96: Emission standard of air pollutants for industrial kiln and furnace
GB 25466-2010: Emission standard of pollutants for lead and zinc industry

References:
Tian, H. Z., Zhu, C. Y., Gao, J. J., Cheng, K., Hao, J. M., Wang, K., Hua, S. B., Wang, Y., and Zhou, J. R.: Quantitative assessment of atmospheric emissions of toxic heavy metals from anthropogenic sources in China: historical trend, spatial distribution, uncertainties, and control policies, Atmos. Chem. Phys., 15, 10127-10147, 2015.
Streets, D. G., Devane, M. K., Lu, Z. F., Bond, T. C., Sunderland, E. M., and Jacob, D. J.:
All-time releases of mercury to the atmosphere from human activities, Environ. Sci. Technol., 45, 10485-10491, 2011.

Compared with previous studies, this updated ISP inventory included consideration of roasting plant and coke oven, counting for 22%-34% of ISP's emissions. I think authors should include some words in introduction to highlight this consideration and stress that these two processes are potentially important in shaping the trends of ISP Hg emissions.
Response: The importance of these two processes was introduced as follows in the introduction.
"In these studies, Hg emissions were determined as the product of crude steel production and unique emission factor of 0.04 g/t steel produced which did not consider the emissions from roasting plant and coke oven (two processes of ISP). However, field experiments in China's ISP indicated that these two processes are significant for Hg emissions. Emissions from coke oven accounted for 17%-49% of the total Hg emissions of ISP (Wang F. Y. et al., 2016). Thus, these two

processes are potentially important in shaping the trends of ISP Hg emissions."
See revised manuscript, Page 3, Line 37-43.

In Wu et al 2016 ES&T paper, authors noted that the Hg emissions from ISP are quite possible to increase, however, in this updated ISP inventory, authors argued that Hg emission from ISP are expected to decrease. I think authors should explain this point clearly and included some discussion in section 3.4.
Response: Revised as follows.
"By using such semi-quantitative method, the emission factor in 2020 (0.0402 g/t for long process and 0.0211 g/t for short process) was almost the same as that in 2015 (0.0403 g/t for long process and 0.0212 g/t for short process). Thus, atmospheric Hg emissions in 2020 will almost depend on crude steel production and the emissions in 2020 will reach 40 t at the conservative situation. Therefore, the technology-based emission factor method will provide more objective forecast of future emissions."
See revised manuscript, Page 15, Line 366-371.

It seems that Hg release rates used in this study (Table S1) are simply averaged by all available data, why?
Response: From the literature collected, we could not find detailed introduction of the furnace as well as the operation conditions. Thus, it is difficult to identify the reasons caused the difference of Hg release rates. Under this situation, we assumed that Hg release rate fit normal distribution and the average mean of Hg release rates for specific process was applied.

Other specific points:
Line70, many studies.
Response: Revised as follows.
"Many studies have reported Hg concentrations in coal (Swaine, 1992; Tian et al., 2010; USGS, 2004; Zhang et al., 2012)"
See revised manuscript, Page 4, Line 70.

L102, "filed experiment" should be "field experiment", some other places should be also revised.
Response: Revised.

L206, Table S3
Response: Revised.
The peak of Hg input was in 2014 when the crude steel production reached the highest value (**Table S3**).
See revised manuscript, Page 9, Line 209.

**Response to reviewer 2**
General comments:
The authors made a great effort on the estimation of Hg emission from iron and steel production (ISP) during 2010-2015 in China by using technically-based emission factor

method with up-to-date parameters through national sampling and literature review, which has reduced the associated uncertainty of Hg emission from ISP. Given the updated Hg emission database, this work provides meaningful results to health-related study and is helpful to make control policy. Frankly speaking, many sections in this paper just reported the numbers but didn't provide further insight beyond those facts.

Response: Discussions were added as follows.

"The slower AAGR of Hg emissions (7.2%) than that of crude steel production (13%) reflected the impact on Hg emission reduction due to energy saving and environmental protection in ISP. On one hand, Hg input to produce unitary crude steel decreased from 0.17 to 0.12 g/t, which mainly benefited from the improvement of coke production efficiency and energy utilization efficiency of sinter/pellet plant and blast furnace. Since 2004, indigenous coke production method with high coal consumption has been gradually replaced with machine coke production method. The coke ratio in sinter/pellet plant has been reduced from approximately 388 kg/t pig iron produced in 2000 to 363 kg/t in 2015 (CISIA, 2001-2016). On the other hand, the improvement of APCDs increased the overall Hg removal efficiency from 47% in 2000 to 65% in 2015 (**Fig. 3**). APCDs for coke oven have shown the largest Hg removal efficiencies (64%-87%) while pollution control in sinter/pellet plant contributed most to the rapid Hg reduction speed during 2000-2015. The replacement of CYC and WS with ESP and FF in sinter/pellet plant improved Hg removal efficiency from 21% in 2000 to 44% in 2010. The application of FGD in addition to dust collectors was the main driver of Hg reduction in sinter/pellet process during 2011-2015. Hg removal efficiency in sinter/pellet plant was 53% in 2015."

See revised manuscript, Page 10, Line 232-247.

"The comparison of emissions from different types of process combination in this study indicated the significance of including emissions from roasting plant and coke oven in the ISP emission inventories. The proportion of emissions from these two processes accounted for 22%-34% of ISP's emissions during the whole study period. In addition, these two processes were important in shaping the trends of ISP Hg emissions. For example, Hg emissions of all processes showed an increasing during 2007-2008 (Red line in Fig. 6). However, if these two processes were not considered, we will observe a decreasing trend (Green and orange line in Fig.6). Moreover, given the impact of APCDs on the emission estimation, inventories in ISP should also apply distrinct APCD profiles for different processes so as to reduce the uncertainty of inventories."

See revised manuscript, Page 13-14, Line 325-334.

Specific comments:

My main concern is the credible of dataset. Does the "SV" in Table 1 means standard deviation value? If it does, how can the authors get SV when they just have one Limestone sample in Shanghai and Gansu?

Response: All collected samples were analyzed in triplicate or more times to obtain parallel results. In Table 1, if only one sample was collected, the standard deviation of the parallel Hg concentrations of this sample was signed as the SD in Table 1. To avoid confusion, we have deleted the data.

See revised manuscript, Page 20, Table 1.

And in Table 2, the emissions of Hg in Hainan in 2000/2005/2010/2015 were 0.0 but it has an average annual growth rate of 26%. Is that because the emission amount of Hg in Hainan was quite low so that it doesn't show the true value when you only keep one decimal fraction?

Response: Yes. To keep the true value of the data, we revised the unit of the data in Table 2 from "t" to "kg".

See revised manuscript, Page 22, Table 2.

In addition to this, what is the "All ISP processes" means in Fig.4? If it means the sum of Hg emission from all processes, why the total emission of Hg (asterisk in Fig.4) in Roasting plant, Coke oven, Sinter/Pellet, Blast furnace, Oxygen and Arc far exceeding the value in All ISP processes, either in 2000 or 2015?

Response: Yes, the all ISP processes means the sum of Hg emissions from all processes. The value in all ISP processes actually equaled to the total emissions of Hg in roasting plant, coke oven, sinter/Pellet, blast furnace, oxygen and arc (Refer to the right coordinate of Fig. 4). The exact data used to plot the graph were listed in the following table.

| Process | Emissions in 2000 (t) | Emissions in 2015 (t) |
|---|---|---|
| All processes | 11.5 | 32.7 |
| Roasting plant | 1.1 | 3.5 |
| coke oven | 2.9 | 3.5 |
| Sinter/Pellet | 4.8 | 15.9 |
| Blast | 1.9 | 7.9 |
| Oxygen | 0.2 | 1.4 |
| Arc | 0.5 | 0.5 |

To better understand Fig 4, we have re-drawn the figure. An arrow was added to point the right coordinate。

[Figure]

Given that the annual Hg emission proportion (by comparing Fig. 2 and Fig. 3) shows a decreasing trend since 2000, I'd suggest authors add the emission proportion values to Figure 3 because it will provide more insight of the impact of energy consumption and air pollution control on Hg emission.

Response: Hg removal efficiency was added to Figure 3 to provide insight of impact of air pollution control.

[Figure]

Other specific points:

Line 323: the authors stated ": : : indicated the significance of our new inventory", which makes me confused. Please specify what it exact means.

Response: We have specified the meaning of this sentence, which was revised as follows.

"The comparison of emissions from different types of process combination in this study indicated the significance of including emissions from roasting plant and coke oven in the ISP emission inventories. The proportion of emissions from these two processes accounted for 22%-34% of ISP's emissions during the whole study period. In addition, these two processes were important in shaping the trends of ISP Hg emissions. For example, Hg emissions of all processes showed an increasing during 2007-2008 (Red line in Fig. 6). However, if these two processes were not considered, we will observe a decreasing trend (Green and orange line in Fig.6). Moreover, given the impact of APCDs on the emission estimation, inventories in ISP should also apply distrinct APCD profiles for different processes so as to reduce the uncertainty of inventories."

See revised manuscript, Page 13-14, Line 325-334.

Line 335: the authors quoted "emissions of pollutants are required to be reduced by 15% for ISP before 2020 in China". That is unclear to me what the pollutants really are? Does the government required to reduce all pollutants by 15% simultaneously?

Response: The pollutants generally referred to those considered in the emission standards of air pollutants for iron and steel industry, including $SO_2$, $NO_x$, PM, dioxin, and fluoride. Yes, the government required that the above pollutants should be reduced by at least 15% at before 2020. Revised as follows.

"On the other aspect, emissions of pollutants (eg., $SO_2$, NOx, and PM) are required to be reduced by at least 15% for ISP before 2020 in China (MIIT, 2016)."

See revised manuscript, Page 14, Line 353-355.

Technical correction:

Line 62, 102, 152 and some other places: "filed" should be "field".

Response: Revised.

Table 1: "NM" in Note 2 should be revised as "NS".

Response: Revised.

See revised manuscript, Page 21, Table 1.

Figure 1: what is the numeric "1" in the title of figure 1 used for?

Response: The numeric"1"is a note for figure 1. To be clearer, the note was added at the end of the figure instead of the title.

[Figure]

Note:1. Some plants roasted limestone and dolomite separately.
2. Mainly coke breeze. Some plants also use coal powder as fuel.
3. The flue gas after dust collectors are collected in gasometer before use.

See revised manuscript, Page 23, Fig. 1.

---

## Author Response (AR2)

Dear editor,

We have revised our manuscript according to your recommendations. The point-to-point responses were listed as follows.

1. Page 13, Line 330, "an increase during …"
Response:Revised.

2. Page 15, Line 364-370. I think this part represents a comparison but is kinda apart from the context. I suggest rephrase the sentences. For example, "… will be reduced to 27t in 2020. This is much lower than the estimation from our previous study using the transformed normal distribution function...".
Response:Revised.
"If we assumed that the crude steel production reached a conservative value of 1000 Mt and that advanced dust collectors (ESP or FF), desulfurization towers, and denitration technologies were fully applied in ISP, atmospheric Hg emissions in ISP will be reduced to 27 t in 2020. Thus, a decreasing trend will be expected from 2015 to 2020. Such conclusion is opposite with the study using transformed normal distribution method (Wu et al., 2016)."
See revised manuscript, Page 14-15, 361-366.

3. Figure 4 has been revised into black/white version as compared with the ACPD version. I think the color one is better.
Response:Revised.

4. Page 15, Line 384-385, "… were the largest two emission processes, accounting for 49% and 24% of national emissions, respectively."
Response:Revised.

5. In table 1, column for NS of Limestone, some numbers "1" are in red, what does it mean?
Response:The red color was used to highlight our revision in previous manuscript. The color was changed.

6. I suggest to add words on the improvement of the method and finding in the conclusion. For example, Line 390 can be extended to specify the innovation and merits of the method and findings compared with previous studies.
Response:Revised.
"In this study, we applied the technology-based emission factor method for better quantification of Hg into ISP and atmospheric Hg emissions from different processes of ISP. Compared with previous studies, the uncertainty of atmospheric Hg emissions from ISP has largely reduced with better understanding of Hg flow in ISP. This method has provided more objective estimation of current emissions and forecast of future emissions."
See revised manuscript, Page 16, 392-396.